# Application of Deep Learning to IVC Filter Detection from CT Scans

**DOI:** 10.3390/diagnostics12102475

**Published:** 2022-10-13

**Authors:** Rahul Gomes, Connor Kamrowski, Pavithra Devy Mohan, Cameron Senor, Jordan Langlois, Joseph Wildenberg

**Affiliations:** 1Department of Computer Science, University of Wisconsin-Eau Claire, Eau Claire, WI 54701, USA; 2Interventional Radiology, Mayo Clinic Health System, Eau Claire, WI 54703, USA

**Keywords:** deep learning, medical imaging, Convolutional Neural Networks, UNet, IVC filter

## Abstract

IVC filters (IVCF) perform an important function in select patients that have venous blood clots. However, they are usually intended to be temporary, and significant delay in removal can have negative health consequences for the patient. Currently, all Interventional Radiology (IR) practices are tasked with tracking patients in whom IVCF are placed. Due to their small size and location deep within the abdomen it is common for patients to forget that they have an IVCF. Therefore, there is a significant delay for a new healthcare provider to become aware of the presence of a filter. Patients may have an abdominopelvic CT scan for many reasons and, fortunately, IVCF are clearly visible on these scans. In this research a deep learning model capable of segmenting IVCF from CT scan slices along the axial plane is developed. The model achieved a Dice score of 0.82 for training over 372 CT scan slices. The segmentation model is then integrated with a prediction algorithm capable of flagging an entire CT scan as having IVCF. The prediction algorithm utilizing the segmentation model achieved a 92.22% accuracy at detecting IVCF in the scans.

## 1. Introduction

Venous thromboembolism (VTE) are blood clots that begin in vein, a common disorder that affects 1 in 1000 people annually [1]. The standard treatment are anticoagulants (blood thinners) which are highly effective at reducing the risks of ongoing and recurrent VTE. However, some patients cannot tolerate anticoagulation. In these cases, the insertion of a IVCF is recommended to reduce the risk of a pulmonary embolism [1,2]. They are a second-line intervention as they do not treat an existing VTE and instead act as prevention against a pulmonary embolism.

Originally developed in a permanent form in the 1970s, a retrievable version was approved for use by the FDA in 2003 [3], while greater than 85% of filters are initially placed with the intention of temporary use, a retrospective review found that up to two-thirds were never retrieved [4]. In theory, not retrieving an IVCF should not be a problem as all filters are FDA-approved for permanent use in addition to temporary indications. Unfortunately, numerous studies have shown the potential for serious complications resulting from long dwell times of filters, with the risk of complication increasing almost linearly with time [5,6]. This was highlighted in an FDA communication, released in August 2010 and updated in May 2014, that recommended retrieval as soon as possible and estimated the risk/benefit begins to favor retrieval between 29 and 54 days after placement unless there was need for prolonged IVC filtration [7]. Importantly, that communication explicitly placed the responsibility of evaluating the ongoing need for filtration on the implanting physician. The communication also had the side effect of drastically increasing IVCF-related litigation [8].

Retrieval of IVCF is important to avoid potential long-term complications that may arise. These complications include filter migration, caval thrombosis, filter fracture and caval penetration [2,9,10]. The risk of these complications occurring increases with longer dwell times [2,10]. However, the study in [4] also suggested that “loss to follow-up” or discontinued care and lack of a tracking program account for at least 20% of the filters that were not retrieved.

It is now basically mandatory that all physician groups who place IVCF, primarily Interventional Radiologists, have a method for tracking patients and removing filters that are no longer indicated. This is commonly done using a spreadsheet, which has the significant limitation that the information is not shared even across physician groups within the same enterprise. As patients transfer their care from one physician group to another it is common to become “lost to follow-up.” Improvements in tracking beyond the simple spreadsheet certainly have many advantages but cannot help identify patients whose filter was placed elsewhere but are now under the purview of a new health system or physician group.

Fortunately, IVCF are readily visible on CTs of the abdomen and patients receive this type of imaging commonly and for reasons unrelated to the filter itself including injury, abdominal pain, screening (e.g., CT colonography instead of a colonoscopy) or cancer staging/surveillance. Radiologists usually comment if a filter has any finding of a complication but often do not make any comment if a filter appears normal. This results in poor reliability of automated natural language processing (NLP) methods to identify previously unknown filters. Given that there are approximately 400 eligible CT scans performed every day within the Mayo enterprise and low prevalence of IVCF in the general population, a detection algorithm with low accuracy would place an untenable burden on clinician review of the inevitable false positive results. Therefore, an algorithm with greater accuracy, and most importantly one with high specificity, is needed.

Within the broader construct of Artificial Intelligence (AI), machine learning algorithms are used to learn patterns, gather statistics on input data and use that information for prediction of future unseen data. The performance and prediction accuracy of these algorithms is dependent on quality and presentation of raw input. The concept of better representation of raw data to the machine learning models is termed as feature engineering [11,12,13,14] and a significant amount of time must be spent in manually creating important features from the existing ones [15,16,17]. Deep learning algorithms, which are a subset of machine learning, offer an advantage in this domain as they can automatically extract relevant features without the requirement of human input using a layered and hierarchical structure [18]. A class of deep learning architecture known as a Convolutional Neural Network (CNN) will be explored in this study due to its ability to accurately classify images by incorporating their spatial aspect [19,20].

A CNN consists of convolutional layers stacked one after another. There are three basic steps to convolution which are repeated several times: convolution, activation, and pooling. In convolution, an image is analyzed by sliding several square grids of a certain pixel size across the entire image. This process is referred to as a sliding window approach and each of these sliding windows are called filters or kernels [21]. Each kernel extracts certain features in the image. The output of a layer is next passed through an activation function introducing non-linearity into the model, which is the one of the biggest strengths of a neural network. Rectified Linear Unit (ReLU) [22] is the most commonly used activation function in this stage. In pooling, an image is analyzed in multiple resolutions to detect low, medium and high-level features. Max-pooling is commonly employed in this stage wherein the images are downscaled multiple times. Since the convolution operation repeats on the image in multiple resolutions, we can extract features from the image with superior accuracy [23].

This project aims to develop a simple yet effective AI-based algorithm with significant accuracy to automatically analyze all CT scans and produce a binary output indicating the presence or absence of an IVCF. Due to the narrow focus and retrospective nature of the task, it can also serve as an example for rapidly developing an AI tool and deploying it without any risk to patients. With almost 811,487 IVCF placed between 2005 and 2012 according to data from the Agency for Healthcare Research and Quality Health Care Utilization Project Nationwide Inpatient Sample, it is imperative that a follow-up system is made to ensure improved patient care [24].

## 2. Artificial Intelligence and Medical Imaging

Recent developments in technology have brought about the digitization of healthcare with up to 84% of general medical hospitals in the United States having adopted the use of electronic health record (EHR) systems as of 2015 [25]. This abundance of digitized data has allowed AI to revolutionize the healthcare industry and improve patient care. One of these revolutionary techniques is the utilization of deep learning models, specifically CNNs, for medical image analysis [26] to diagnose medical conditions.

Image segmentation and classification tasks are of especially high priority in the medical imaging field [1]. CNNs are a prime tool for these tasks due to its ability to learn highly discriminative features present in the images [27,28]. UNet is a type of CNN architecture which was initially developed for biomedical image segmentation purpose. This architecture outperformed other image segmentation methods of its times including the previous best method [29]. Since then, variants of the UNet architecture have been used extensively for medical image analysis. This includes but is not limited to diagnosing cancer (liver, lung, cervical, etc.) as well as segmenting hard and soft tissues in CT scans [30]. UNet architecture was also used to segment the IVC lumen in intracardiac echocardiography images as part of a proposed pipeline for image-guided vascular navigation [31]. CNNs have successfully been used to detect the presence of lung cancer in CT scans. In [32], authors suggested that with appropriate preprocessing steps, CNNs are able to classify CT scans as positive or negative for lung cancer with high accuracy. Similarly, Ref. [33] proposed a two-module network to detect the presence of malignant pulmonary nodules. The first module is a 3D-region proposal network that detects suspicious nodules in the scan while the second module evaluates the cancer probability based on the most suspicious nodules using a leaky noisy-or model. CNNs have also been used extensively for detection of tuberculosis [20,34,35], cancer detection [36,37,38,39,40] as well as COVID-19 [41,42,43,44]. Domain adaptation using unsupervised machine learning have also been successfully applied for knowledge extraction and organ segmentation [45,46,47].

In regards to medical image analysis involving IVCF, research has previously been done on classifying the type of IVCF present in radiographs using CNNs. One research focused on classifying 14 different IVCF types in manually cropped radiographic images. A 50-layer ResNet architecture was used with a modified final fully connected layer to perform this classification [48]. Another research attempted to build on this approach by creating an architecture that is able to classify three different IVCF types without the need to crop and manipulate the radiographs to be centered on the filter [49]. Prior research by Dr. Wildenberg, developed a purely image-processing algorithm that can detect a filter from CT scans with a sensitivity and specificity of about 80% each. Given the approximately 400 eligible CT scans performed every day within the Mayo enterprise and low prevalence of IVCF in the general population, this accuracy would place an untenable burden on clinician review of the inevitable false positive results. An automated deep learning approach provides a feasible solution and to our knowledge the domain of real-time IVCF presence/absence detection from CT scans remains unexplored.

## 3. Materials and Methods

The schematic diagram of the proposed approach is shown in Figure 1. It consists of an image augmentation phase, a deep learning model phase, followed by a prediction phase utilizing the deep learning models.

### 3.1. Dataset

Institutional Review Board (IRB) approval was obtained for this project. Approximately 100 known positive (IVCF present) and 100 known normal CT scans were provided by Dr. Wildenberg using the internal list of known patients with filters out of which 90 IVCF and 90 normal scans were finally used. These scans were anonymized by removing any protected health information (PHI) attached to the scan. Slice thickness and width distribution is shown in Figure 2. Initial preprocessing of the CT images was guided utilizing *a priori* knowledge about IVCF (e.g., composed of metal, located approximately within the center of the body, small relative to other structures) to maximize the relevance of the data submitted to the CNN.

A total of 465 CT image slices were used during the training and validation phase from the 90 CT scans with IVCF. These slices had visible IVCF in them. Segmentation of IVCF pixels were generated by the research group under supervision of Dr. Wildenberg and Dr. Gomes to ensure they represent a proper mask for the CNN to train on. CNN development, training, and validation was then performed. The most significant aspect of this project involved exploration of the exact architecture and parameters to yield a high-performing CNN for filter recognition.

The set of 90 normal scans were later used in the IVCF prediction algorithm to validate the performance of the model in the realm of false positives. These scans were not used during the segmentation phase.

### 3.2. Spatial Cropping

Data augmentation both pre-training and during training were conducted with diverse objectives. Prior to training, all scans were spatially cropped to remove as much of background possible giving the deep learning model significantly less extraneous information for better performance. This spatial cropping of 20% was applied across all four edges. Hence the original CT scans of 512×512 were reduced to 307×307 followed by resampling to 256×256. The spatial cropping was followed by removal of CT image slices per scan that were 40 cm below the craniocaudal region. The choice of 40 cm as a cut-off was ideal since nearly all IVCF are located inside that region. Figure 3 shows the distribution of slices per CT scan before and after this reduction. The 40 cm cut-off was able to reduce the total number of CT image slices by 19.01%.

### 3.3. Normalization

Two variants of the IVCF dataset were generated with and without intensity normalization. For soft normalization scheme, CT images were normalized using minimum and maximum Hounsfield Units (HU) values. For Hard Normalization, the maximum and minimum HU values were set to 1 and 2500, respectively. The purpose of this preprocessing step was to ascertain if window width increases the contrast of IVCF making them more prominent for the CNN models. Figure 4 shows the result of the application of the aforementioned intensity normalization.

### 3.4. Image Augmentation

Data augmentation during the training phase satisfied the removal of training bias. Any machine learning model is sensitive to location and orientation of features. As such, the proposed model performed cropping, flipping, and rotation. CT scan and the corresponding label were first resized by a certain margin before applying a random cropping function. This random cropping simulated a patch based analysis where the CNN would randomly extract a subset of the entire CT image slice. This ensured that the model would not receive the same image while training in each epoch. The CT image slices were also randomly flipped from left to right to simulate differences in horizontal orientation of CT image acquisition. These steps ensure that CNN predictions stay robust even if they are used across multiple scanning platforms with different settings. To ensure that image labels are consistent in the resampling process, Nearest Neighbor Interpolation was used compared to Bilinear interpolation for the CT scan images. An example of image augmentation is shown in Figure 5.

### 3.5. Network Architecture

A modified version of UNet was used as a base architecture for this research. This framework was built using online resources available from TensorFlow [50]. The inception of deep learning and image segmentation began with the UNet architecture proposed by Ronneberger et al. [29]. Here each CT image slice is analyzed using a combination of image filters (also known as kernels), and in multiple resolutions. The kernels applied over the image are responsible for identifying hidden pattern in the dataset. Proper application of these kernels including their sizes and dilation rate can have a significant impact on accuracy as well as optimization features [51,52]. For example, a kernel that could be used to delineate edges in the image is the Sobel filter [53]. These filters act as weights during training process allowing the deep learning model to train itself and identify patterns associated with the IVCF. The non-linear approach significantly boosts model performance over traditional machine learning algorithms. The hierarchical structure of the UNet architecture is made possible by modifying the image resolution. Application of kernels at multiple resolutions further allows extraction of diverse features associated with the IVCF. Thus, the underlying architecture can process complex data and extract relevant features in different levels of abstraction.

### 3.6. Training Parameters

The underlying framework of this architecture is visible in Figure 6. Approximately 25 million parameters were used for training. Adam optimizer was used for gradient descent. Leaky Relu activation was used in the downsampling phase to allow negative gradients and further optimize weights for better training. The model was trained for 500 epochs using a batch size of 20. Training and validation accuracy were recorded with 80% of the total images (i.e., 372 slices) being used for training and 20% for validation (i.e., 93 slices). Since this is a segmentation, Sparse Categorical Crossentropy loss was used. The training loss decreased steadily from 0.3276 to 0.0016 for hard normalization scheme. The training loss also decreased steadily from 0.1651 to 0.0015 for soft normalization scheme. Since dropout is not an effective approach in convolution networks [54], it was only applied in the first upsampling layer with a value of 0.5. Batch normalization was used regularly after convolution layers. Deep learning models are computationally intensive. Hence all processing was carried out using the BOSE cluster made available through the Blugold Center for High Performance Computing. The Graphics Processing Unit for this research was NVIDIA Tesla V100S with 32 GB memory.

### 3.7. IVCF Prediction Pipeline

The proposed UNet architecture above is used to develop a model that is able to successfully segment IVCF pixels in the slices of a CT scan. To integrate the tool and enable rapid diagnosis, an IVCF prediction algorithm is proposed. This algorithm generates a comprehensive report for the clinicians about the IVCF and its location. The flagging algorithm should have the potential to reduce false positives that may arise from bone or surgical implants which share similar HU values with IVCF. It uses two parameters (a) *sig_count* and (b) *sequence* to flag a CT scan as having IVCF. Using *sig_count* the clinician sets a threshold for the number of pixels that may belong to an IVCF per CT image slice. The *sequence* parameter takes a spatial approach by looking at how many consecutive slices exceed the *sig_count*. For example, if the *sig_count* = 200 and *sequence* = 5, the prediction pipeline will alert the clinician when five consecutive CT slices have 200 IVCF segmented pixels or more. The clinician can then review the results to see if any IVCF is present. The clinician can also modify the *sig_count* and *sequence* values to fine tune the prediction outputs based on patient and CT scanner characteristics. For example, if a patient has surgical implants, the clinician may feel comfortable increasing these parameters to reduce detection of other devices placed in the body. If the prediction does not satisfy any of the two parameters, the CT scan will be classified as having no IVCF.

The process begins by accepting the segmented prediction from UNet and passing it through scikit-image image processing library in Python [55]. Region properties of the segments are analyzed to remove any spurious segmentation that may arise occasionally. The sub-section of the algorithm for this prediction pipeline is shown in Algorithm 1. The algorithm takes in as input the *images, labels, sig_count*, and *sequence*. The *images* represents a processed NumPy array of a CT scan of a patient, while the *labels* contain a predicted mask for that CT scan. Both these arrays have a dimension of (slices×width×height). The default dimensions of the processed NumPy version of a CT scan used for prediction is (128×256×256). The algorithm also calls a separate function called *displayImage* which is responsible for printing the scans and segmentation masks of the IVCF sequence. The combination of *sequence* and *sig_count* plays an excellent role in IVCF detection and will be discussed in the next section.
**Algorithm 1: **IVCF prediction pipeline using the deep learning model
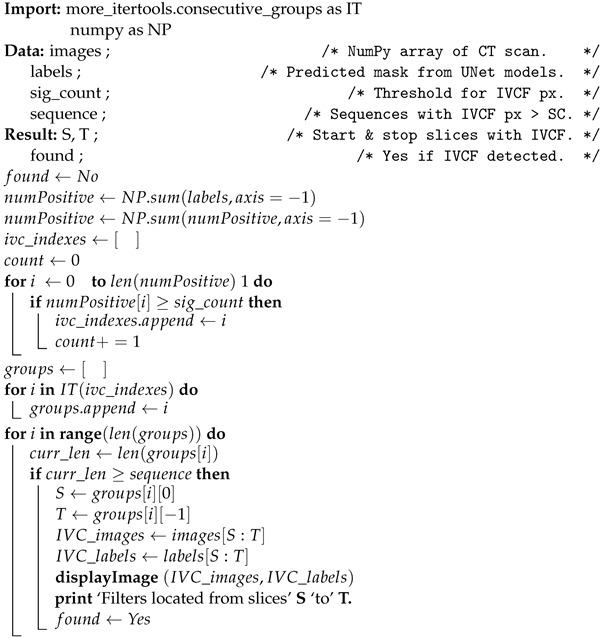


## 4. Results

### 4.1. Segmentation Evaluation

Two separate UNet variants were evaluated. Table 1 highlights the data variants used for training. These variants differ in the hard and soft normalization techniques as highlighted in Figure 4. The mean average precision determines the accuracy between ground truth segmentation mask compared to the prediction masks of IVCF pixels. The Dice score was used to analyze the mean average precision. An example of Dice metric analysis is shown in Figure 7. Here one rectangle references the segmentation and the other is for prediction.

The first variant of the UNet model performed significantly well with Dice score of 0.817 for IVCF on training data and 0.798 for validation data. The number of false positives were significantly low for both training and validation IVCF pixels. During training, only 7.97% of IVCF pixels were misclassified as background and 0.12% of background pixels were classified as IVCF. The training sensitivity was about 0.92 and specificity was almost 0.99. For the hard normalized data, a SegNet variant of the model was also used for comparative analysis. We noticed slightly lower Dice scores at 0.743 for training data. This may be due to inherent architecture of SegNet [56] which only transfers max-pooling indices from the encoder phase. Due to this performance, UNet model was selected in the final prediction pipeline.

The second variant of the UNet model utilized the soft normalized data and followed a similar pattern of experiments as the first variant. The Dice score for IVCF was 0.71 for training and 0.70 for validation data. Here the Dice scores were lower than the first variant. Furthermore, 17.26% of the total IVCF pixels were classified as background and false positives was around 0.15% of the background pixels.

Figure 8 is a sample prediction of CT image slices with IVCF. Column two shows the hard normalized segmentation followed by column three with the soft normalized segmentation outcome. As it can be observed, the segmentation of hard and soft normalization are very comparable to each other. Due to the better Dice scores for hard normalization, it was selected to be used in the IVCF prediction pipeline.

Since the binary segmentation is highly imbalanced, precision-recall curves were generated over ROC plots. These results shown in Figure 9. Comparing the Dice scores for soft against hard normalization revealed that hard normalization is able to help the UNet model to attain precision in its segmentation approach.

### 4.2. IVCF Prediction Pipeline Evaluation

The prediction algorithm was then used on the segmentation output from hard normalized CT scans. Using Algorithm 1, several variations of *sig_count* and *sequence* were used to test the IVCF classification accuracy on all 90 CT scans with IVCF. The combination of three of the most significant parameters are shown in Table 2. Confusion matrices in Figure 10 represents this information as true positives. We noticed that increasing the number of sequential CT slices reduces the classification accuracy of the prediction algorithm. This is mostly due to scanner heterogeneity as some scans with a higher resolution show more of the IVCF profile. A 92.22% detection accuracy from the 90 IVCF scans was observed using *sig_count* = 300 IVCF pixels and *sequence* = 7.

A validation test was conducted using 90 normal CT scans that were not used during segmentation phase. We noticed some limitations with regards to false positives as some of the scans were classified as IVCF. Confusion matrices in Figure 10 show that the highest number of false positives were 13 when using a *sequence* of 5 but only 3 when using *sequence* of 9. Upon closer inspection it was observed that these scans were associated with calcification around spinal foramen region near the IVCF. Since it assumes a ring-like appearance the segmentation model would mislabel the pixels as IVCF. This limitation requires further investigation. We also noticed superior performance of the segmentation model as none of bone or surgical implants were segmented as false positives even though they share similar signature as the IVCF.

To summarize, results from the proposed classifier indicates that the optimum combination for the current dataset is *sig_count* = 300 IVCF pixels and *sequence* = 7 sequential slices in a CT scan. Using this combination produces we achieved a 91.67% overall accuracy using 180 CT scans. Only 8 normal scans are flagged as patients having IVCF. A detailed metric from this analysis is shown in Table 2. Other combinations of *sig_count* and *sequence* were also explored to validate the efficacy of the proposed approach as shown in Table A2. It was noticeable that decreasing the *sig_count* to 200 and *sequence* to 5 produced a 98.8% accuracy in detecting IVCF in scans. However, it also increased the number of false positives.

## 5. Conclusions

In this research we have used concepts of deep learning to create an IVCF detection pipeline. Two UNet models were trained to identify the foundation of our segmentation approach. A prediction algorithm for flagging CT scans with IVCF was constructed. The pipeline returned promising results with prediction time per scan being as low as 43 seconds. Future work will include training the model on a much larger dataset to investigate the possibility of reducing false positives caused by calcification around the spine. The feasibility of model deployment in a testing phase is being explored at Mayo Clinic. Preliminary results show promise for future improvements.

## Figures and Tables

**Figure 1 diagnostics-12-02475-f001:**
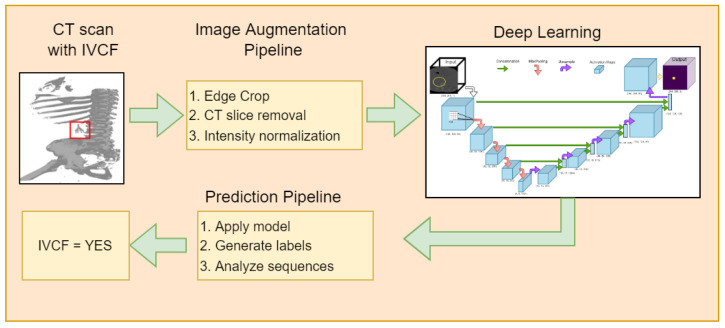
Schematic diagram of the proposed approach to IVCF detection.

**Figure 2 diagnostics-12-02475-f002:**
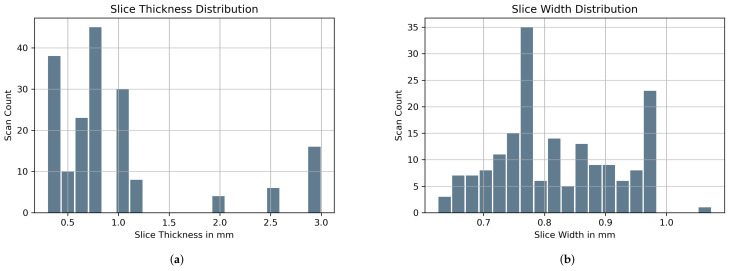
Spatial resolution of 180 CT scans used in the research where (**a**) Slice Thickness and (**b**) Slice Width.

**Figure 3 diagnostics-12-02475-f003:**
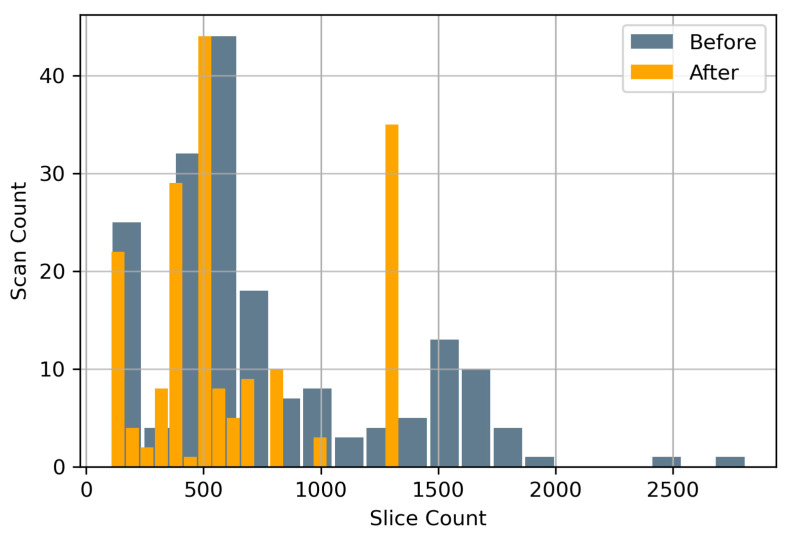
CT Image Slice Count Distribution before and after removal of scans 40 cm below the craniocaudal region.

**Figure 4 diagnostics-12-02475-f004:**
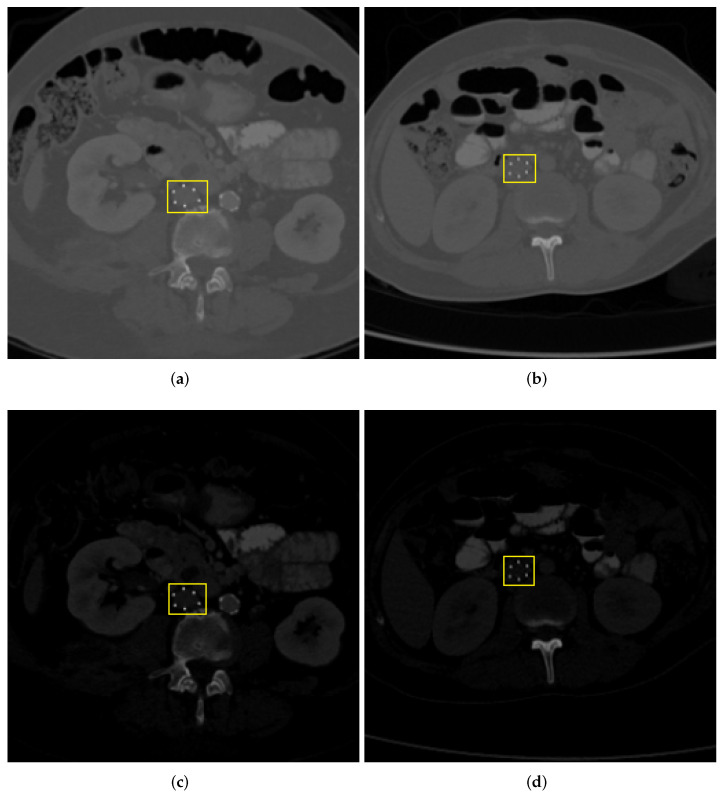
Example of CT scan slices along the axial plane with IVCF. Top row (**a**,**b**) are without application of intensity adjustment (Soft Normalization). Scans (**c**,**d**) have HU values adjusted by setting minimum as 1 and maximum as 2500 (Hard Normalization).

**Figure 5 diagnostics-12-02475-f005:**
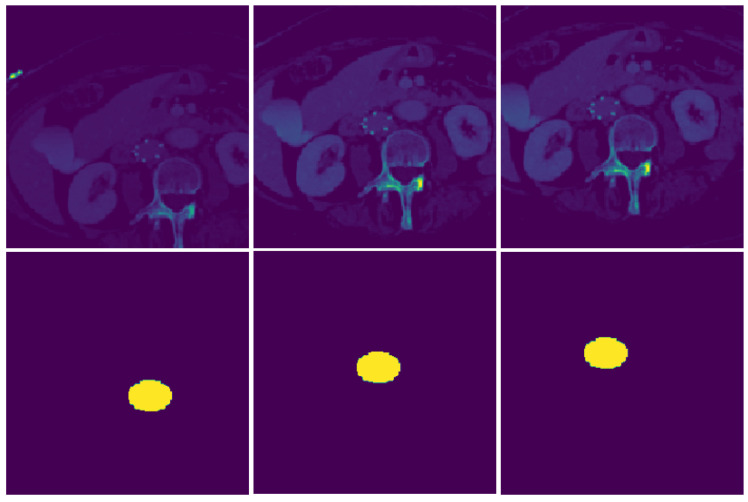
Image augmentation during training process for a CT image slice. These augmentations like random cropping, rotation, and flip reduces spatial bias introduced by the dataset. The masks also undergo similar modifications during training.

**Figure 6 diagnostics-12-02475-f006:**
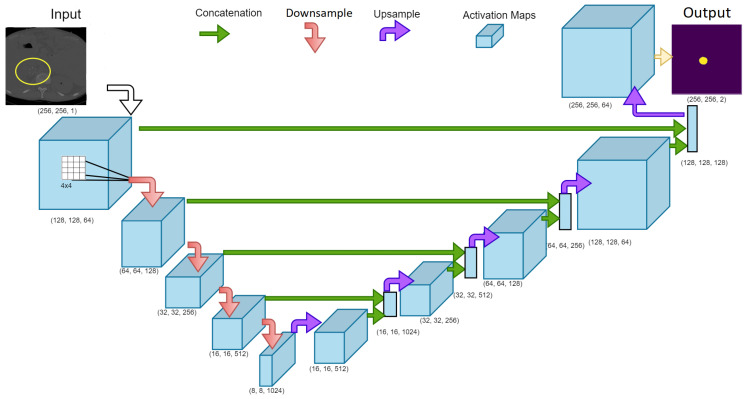
UNet model architecture used for the training phase to generate a segmented map of the IVCF in the CT image slices. The downsampling and upsampling phase were evaluated as separate functions and called using the gradient tape function in TensorFlow. Numbers indicate the (width, height, kernels) used in the layer. Arrows indicate the operation performed while training.

**Figure 7 diagnostics-12-02475-f007:**
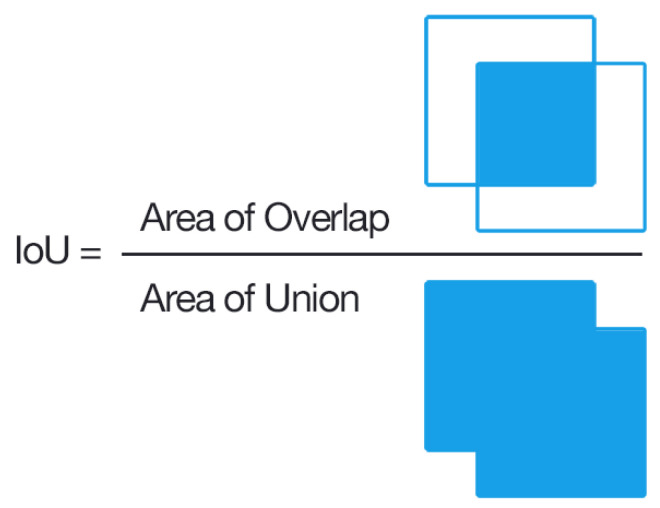
IoU metric evaluation (Dice Score) for IVCF pixels in the CT images. Image by Adrian Rosebrock, distributed under a CC BY-SA 4.0 license.

**Figure 8 diagnostics-12-02475-f008:**
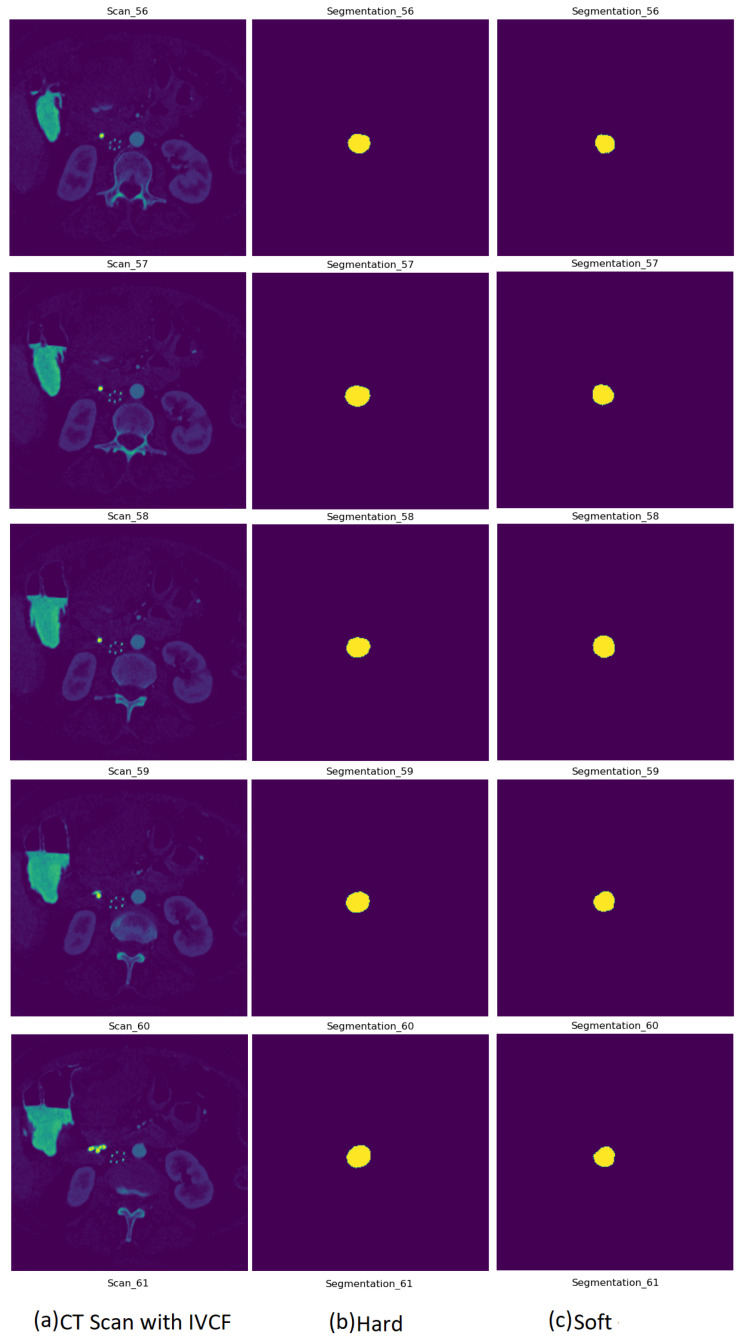
Segmentation output from UNet model variants on a CT scan using hard normalized data.

**Figure 9 diagnostics-12-02475-f009:**
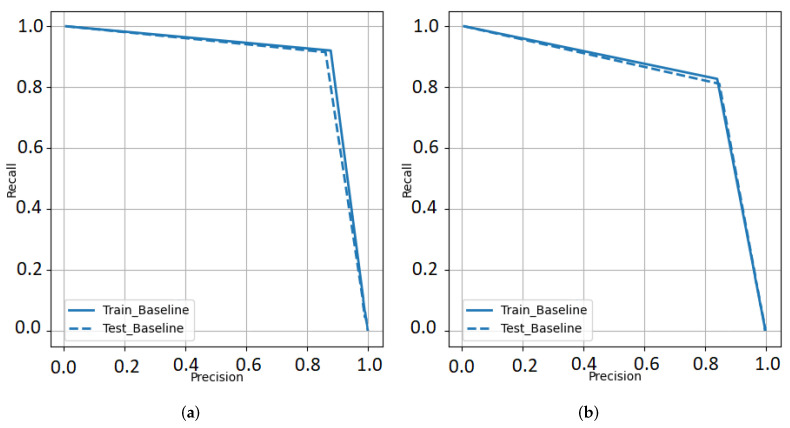
Precision-Recall curves for UNet segmentation approach using (**a**) UNet Model 1 and (**b**) UNet Model 2.

**Figure 10 diagnostics-12-02475-f010:**
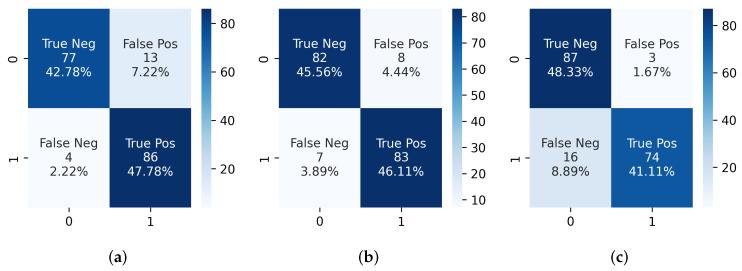
Confusion matrices obtained from IVCF prediction algorithm using (**a**) *sig_count* = 300, *sequence* = 5, (**b**) *sig_count* = 300, *sequence* = 7 and (**c**) *sig_count* = 300, *sequence* = 9. Here 1 represents IVCF scan and 0 represents normal scan.

**Table 1 diagnostics-12-02475-t001:** Dice score for different model variants.

UNet Model	Normalize	Dataset	Dice Scores	
			Background	IVCF
1	Hard	Training	0.9981	0.8168
		Validation	0.9979	0.7981
2	Soft	Training	0.9969	0.7153
		Validation	0.9970	0.7082

**Table 2 diagnostics-12-02475-t002:** Prediction pipeline performance for IVCF detection metrics from Figure 9.

Combination	Accuracy	Sensitivity	Specificity	Precision	F1 Score
300, 5	90.56%	0.9556	0.8556	0.8687	0.9101
300, 7	91.67%	0.92	0.9111	0.9121	0.9171
300, 9	89.44%	0.8222	0.9667	0.961	0.8862

## Data Availability

Links to our scripts used for analysis can be found here https://github.com/rahulgomes19/IVC_2D (accessed on 14 September 2022).

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
