# Peer review of "Application of Deep Learning to IVC Filter Detection from CT Scans"

_diagnostics, 2022, doi:10.3390/diagnostics12102475_

Round 1

Reviewer 1 Report

This study focused on the detection of IVC filters from CT scans. All the materials provided in the manuscript are good. However, I would like to advise the authors to emphasize the reasons why must use the deep network on this issue. Since the shape of IVCFs are regular, it is supposed not to be too difficult to segment the six-point object from the CT slides by traditional image process techniques. By suitable normalization in intensity of the CT slides and its size, an IVCF could be easily found on the slide picture. Is there any necessary reason to detect this object by neural networks? Would the authors provide a reference report on the comparison or put a comparison in this paper between the results by the deep learning method and the traditional image process?

    For the detection results, the authors are advised to put more discussion on possible ways to promote the accuracy for the comfusion matrices in Fig. 10. The authors can try different training parameters to achieve different true positive percentages. Which would be the optimal results? Will a high false positive be little harm or even acceptable for practical investigation? As an auxiliary tool for detecting an object, in my view point, a high true positive rate even with a high false positive rate may be preferred in practices. Therefore, the authors need to assure that it was the optimal results in Fig. 10.

    In the description of training parameters in sec. 3.6, more details would be preferred especially the hyper parameters, the use of early stop, pruning, and the evolution of loss function.

Author Response

Thank you for the feedback. Our responses are attached. 

Reviewer 2 Report

This paper aims at detecting IVC filters by using deep learning method, which has good clinical application value. The results look promising. However, the paper may have the following problems, which can be improved.

Q1: There are some spelling and grammatical errors, please proofread the paper carefully.

Q2: Several recent literatures related to this work are not cited, as follows:

[1] Source-free unsupervised domain adaptation for cross-modality abdominal multi-organ segmentation. Knowledge-Based Systems, 2022, 109155. DOI: 10.1016/j.knosys.2022.109155

[2] Unsupervised domain adaptation for cross-modality liver segmentation via joint adversarial learning and self-learning. Applied Soft Computing, 2022, 121: 108729. DOI: 10.1016/j.asoc.2022.108729

[3] Improvement of cerebral microbleeds detection based on discriminative feature learning. Fundamenta Informaticae, 2019, 168(2-4): 231-248. DOI: 10.3233/FI-2019-1830

Q3: What is dynamic augmentation? What is the difference between this method and the traditional data augmentation? What are the advantages of this method?

Q4: The input of Figure 6 is a two-dimensional image, which does not need to be represented by a cuboid, and will mislead the reader.

Q5: The legend in Figure 9 should not obscure the curve.

Q6: The number of slices with and without IVCF in the training data set and verification data set should be described.

Q7: In this paper, the segmentation effect of unet on IVCF in the training dataset is not good, whether it is possible that the network model is not deep enough, the number of filters is not enough, or the network itself has limited performance. Please provide ablation experiments for selecting network parameters, and compare unet with other classic semantic segmentation networks, such as segnet, FCN, etc

Author Response

Thank you for the feedback. Attached are the responses. 

Round 2

Reviewer 2 Report

The authors have been addressed my questions